# GRADIENT REGULARIZATION IMPROVES ACCURACY OF DISCRIMINATIVE MODELS

**Dániel Varga, Adrián Csiszárik, Zsolt Zombori**
Alfréd Rényi Institute of Mathematics
Hungarian Academy of Sciences
Budapest, Hungary
`{daniel,csadrian,zombori}@renyi.hu`

## ABSTRACT

Regularizing the gradient norm of the output of a neural network with respect to its inputs is a powerful technique, rediscovered several times. This paper presents evidence that gradient regularization can consistently improve classification accuracy on vision tasks, using modern deep neural networks, especially when the amount of training data is small. We introduce our regularizers as members of a broader class of Jacobian-based regularizers. We demonstrate empirically on real and synthetic data that the learning process leads to gradients controlled beyond the training points, and results in solutions that generalize well.

## 1 INTRODUCTION

Regularizing the gradient norm of a neural network's output with respect to its inputs is an old idea, going back to *Double Backpropagation* (Drucker & LeCun, 1991). Variants of this core idea has been independently rediscovered several times since 1991 (Sokolic et al., 2017; Ororbia II et al., 2017; Czarnecki et al., 2017; Gulrajani et al., 2017), most recently by the authors of this paper. Most recent applications (Gu & Rigazio, 2014; Sokolic et al., 2017; Ororbia II et al., 2017; Slavin Ross & Doshi-Velez, 2017) focus on robustness against adversarial sampling (Szegedy et al., 2013). Here we argue that gradient regularization can be used for the more fundamental task of increasing classification accuracy, especially when the available training set is small. Calculating and implementing gradient regularization terms is made easy and fast by modern tensor libraries, making our proposed approach readily available for networks that are hundreds of layers deep.

Our work has explored a broad class of of Jacobian-based regularizers, which provides a unified framework for various gradient regularization approaches. In this extended abstract we present the two most promising variants: 1) classic *Double Backpropagation* (Drucker & LeCun, 1991) which we refer to as *DataGrad* after Ororbia II et al. (2017) who discovered it independently, and 2) *SpectReg* which is our contribution. Our complete results can be found in Varga et al. (2017).

## 2 ANALYSIS

We consider feed-forward classifier networks with a loss function $L(x, y, \Theta) = M(f(x, \Theta), y) = M(\text{softmax}(g(x, \Theta)), y)$, where $x$ is the input, $y$ is the one-hot encoded desired output, $f$ represents the network with a $\text{softmax}$ layer on top, $\Theta$ are the network parameters and $M$ is the categorical cross entropy function. The inputs and outputs of the $\text{softmax}$ layer are called *logits* and *probabilities*, respectively. The central object of our investigation is the Jacobian of the logits $J_g(x) = \frac{\partial}{\partial x} g(x)$ with respect to the inputs.

Gradient regularization penalizes large changes in the output of some neural network layer, to enforce a smoothness prior. We get different variants depending on where the gradients are computed (logits, probabilities, loss term), with respect to what (inputs or some hidden activations) [1], what loss function is used to create a scalar loss.[2] Some of these variants require the expensive computation of

---

[1] Controlling changes with respect to hidden activations is a promising future direction.

[2] In our investigation, we use the squared $L2$ norm, but the $L1$ norm could also be reasonable.

a full Jacobi matrix, which can be made efficient by the application of some projection, introducing new variants.

The two variants presented in this extended abstract are

- **DataGrad** (Double Backpropagation): penalize the $L2$ norm of the gradient of the original loss term with respect to the inputs.

$$L_{DG}(x, y, \Theta) = L(x, y, \Theta) + \lambda \|(\frac{\partial}{\partial x} L(x, y, \Theta))\|_2$$

- **SpectReg** (Spectral Regularization): penalize the $L2$ norm of the randomly projected Jacobian of the logits with respect to the inputs.

$$L_{SpectReg}(x, y, \Theta) = L(x, y, \Theta) + \lambda \|P_{rnd}(J_g)\|$$

where $P_{rnd}(J_g) = J_g^T r$ and $r \in \mathcal{N}(0, I^m)$, and $m$ is the number of class labels.

These variants are more related than it might first appear. As we prove it in Varga et al. (2017), DataGrad as well can be interpreted as minimizing the $L2$ norm of a particular projection of the Jacobian of the logits. This projection is $P_{DataGrad}(J_g) = J_g^T(f(x) - y)$, determined by how well the model predicts the desired output. In contrast, SpectReg controls the Jacobian in all directions.

While both DataGrad and SpectReg work with the Jacobian $J_g$, they avoid having to compute the full matrix. The trick that achieves this is based on the linearity of the gradient. If our regularizer is of the form $\|J_g^T w\|^2$, where $w$ is a vector from logit space, then $\|(\frac{\partial}{\partial x} g(x))^T w\| = \|\frac{\partial}{\partial x} \langle g(x), w \rangle\|$. Hence, expensive Jacobian calculation can be replaced with a gradient calculation. Besides speeding up computation, this can also introduce beneficial regularization. The idea of applying a random projection on the Jacobian also appears in Czarnecki et al. (2017).

If the random normal projector of SpectReg is normalized onto the unit sphere (*spherical SpectReg*), the norm of the projection is a lower bound to the (hard to compute) spectral norm of the Jacobian, which motivates our naming. Furthermore, one can easily show that spherical SpectReg is an unbiased estimator of the Frobenius norm of the Jacobian. Consequently, so is SpectReg, up to a constant scaling. We have not observed any empirical differences between the spherical and the unnormalized variants. For multi-valued functions, directly calculating, or even approximating the spectral norm is infeasible. The Frobenius norm is within a constant factor of the spectral norm, so it can be interpreted as a proxy when our goal is to enforce a Lipschitz property locally.[3]

It is instructive to consider the toy edge case when the neural network consists of a single dense linear layer. Here the weight matrix and the Jacobian coincide. Thus, minimizing the Frobenius norm of the Jacobian coincides with weight decay. The Frobenius norm is submultiplicative, and the gradient of the ReLU is upper bounded by 1. Thus, for a dense ReLU network the product of layer-wise weight norms is an upper bound for the Frobenius norm of the Jacobian. Applying the inequality of arithmetic and geometric means, we can see that the Frobenius norm can be upper bounded by the total weight norm. This suggests an inherent connection between weight decay and gradient regularization, which is worth further investigation.

A reasonable objection to gradient regularization methods is that they control the gradients only in the training points. A highly over-parameterized network is capable of representing a "step function" that is extremely flat around the training points and contains unwanted sudden jumps elsewhere. All our experiments indicate, however, that stochastic gradient descent does not reach these pathological minima and the learned function has smaller gradient norms in randomly selected test points as well. We are still working to better understand this phenomenon, and we believe it can lead to important insights into the learning process.

## 3 EXPERIMENTS

We present a selection of experiments conducted on the MNIST and CIFAR-10 datasets. More results and more details can be found in Varga et al. (2017). We find that gradient regularization increases classification accuracy in a wide range of scenarios, compared with strong baseline models.

---

[3]However, the example of L1 and L2 weight regularization reminds us that optimizing different regularization terms can lead to very different behavior even when they are within a constant factor of each other.

**Gradient Regularization vs. Dropout and Batch Normalization** On MNIST, both gradient regularizers outperform Dropout (Srivastava et al., 2014) and Batchnorm (Ioffe & Szegedy, 2015). Combining DataGrad and Dropout provides the best test accuracy. Table 1 summarizes these results.

Table 1: Comparison of Dropout, Batch Normalization, DataGrad and SpectReg, on MNIST with training size 2000. Numbers are averages of 10 runs, with standard deviations in parentheses.

|  | NoGR | SpectReg | DataGrad |
|---|---|---|---|
| Baseline | 96.99 (0.15) | 97.59 (0.13) | 97.56 (0.24) |
| Batchnorm | 96.89 (0.23) | 96.94 (0.27) | 96.89 (0.22) |
| Dropout | 97.29 (0.19) | 97.65 (0.14) | **97.98** (0.12) |

**Gradient Regularization on a residual network alongside data augmentation** Data augmentation is a crucial ingredient for building models with good generalization properties. The role of standard regularization methods to prevent overfitting often diminishes when used alongside data augmentation as reported by Pereyra et al. (2017). Using a well tuned baseline on the augmented full CIFAR-10 dataset that achieves $93.71\%$ accuracy, we obtain small improvement with SpectReg $(93.74\%)$ and significant improvement with DataGrad $(94.14\%)$.

**The effect of training set size** The effect of regularizers is more significant for smaller training sets, however, we show in Figure 1 **Left** that they maintain a significant benefit even for as much as 20000 training points on MNIST. Besides DataGrad and SpectReg, we also compare *Confidence Penalty* by Pereyra et al. (2017), *Jacobian Regularizer (JacReg)* by Sokolic et al. (2017), *FrobReg* (which directly minimizes the Frobenius norm of the Jacobian without projection) and a baseline model with weight decay. We find that DataGrad performs better than any of its peers for all sizes.

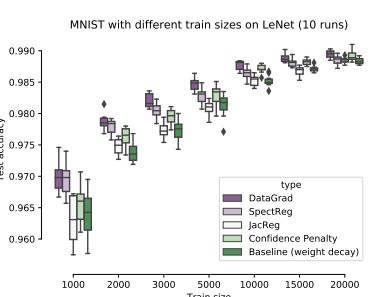 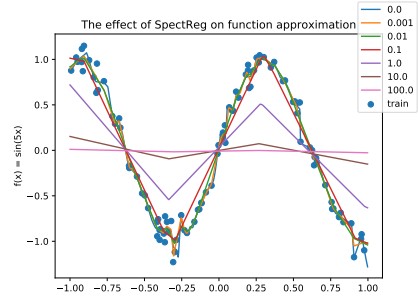

Figure 1: **Left:** Comparison of various regularization methods on MNIST using different training set sizes. DataGrad performs best consistently for all sizes. **Right:** Increasing the weight of the SpectReg regularizer forces the network to learn an increasingly flat function. Although the gradient is controlled only in 100 training points, the whole manifold becomes smoother.

**Local gradient control does not lead to pathological gradient landscape** Using a small synthetic dataset generated from $f : \mathbb{R} \to \mathbb{R}, f(x) = \sin(5x)$, we show in Figure 1 **Right** that SpectReg [4] makes the output smoother, not only around the training points, but globally as well.

## 4 CONCLUSION

Our paper presents evidence that gradient regularization can increase classification accuracy in vision tasks. We identify two methods that outperform strong baselines: *DataGrad* and *Spectral Regularization*. The improvement is most pronounced for smaller training set sizes. Despite the fact that gradient control is applied only at the training points, we find that stochastic gradient descent converges to a solution where gradients are globally controlled. Even for very small training set sizes, the regularized models become smoother on the whole data manifold.

---

[4]We obtain similar curves when using DataGrad.

## ACKNOWLEDGEMENT

The research leading to these results has received funding from the European Research Council under the European Union's Seventh Framework Programme (FP7/2007-2013) / ERC grant agreement 617747. The research was also supported by the MTA Rényi Institute Lendület Limits of Structures Research Group. We thank Csaba Szepesvári and Christian Szegedy for helpful discussions.

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
