# OpenReview forum: "Gradient Regularization Improves Accuracy of Discriminative Models"
_ICLR.cc/2018/Workshop — Reject_

### Official Review · AnonReviewer2 · 2018-03-07
**Experiments Need Improvement**

**Rating:** 5
**Confidence:** 4

**Review:**

This paper compares two gradient regularizers, i.e., (1) penalizing the 2-norm of gradient of loss w.r.t. input and (2) penalizing the 2-norm of randomly projected Jacobian of logits w.r.t. the input. Authors did experiments on MNIST and CIFAR10 to show that gradient regularizers help improve the generalization performance.

My main concerns are as below:
(1) The experimental results are not impressive at all. First, the improvement on MNIST does not tell much about the true story since the baseline itself is worse than expectation (a vanilla CNN can achieve 1.19% error rate whereas yours is 3.01%).
Second, on CIFAR10, the improvement of using SpectReg (0.03%) is negligible. According to my experience, the variance introduced by random seed on CIFAR10 can sometimes lead to change of performance as large as 0.5%.  I recommend authors run the experiments multiple times and report the average for a fair comparison.
(2) In section 2, authors claim "spherical SpectReg is an unbiased estimator of the Frobenius norm of the Jacobian". To be more precise, I think the statement should be "squared spherical SpectReg is an unbiased estimator of the squared Frobenius norm of the Jacobian".

---

> ### Author Response · Authors · 2018-03-21
> **Please consider that our MNIST experiments only used 2000 training points**
>
> "a vanilla CNN can achieve 1.19% error rate whereas yours is 3.01%"
> We are not aware of such a claim in the literature, and would be grateful for a reference. Note that training happened on 2000 randomly sampled data points, rather than the standard 50000 data points.
>
> Thank you for spotting the misstatement in Section 2!
>
> "Second, on CIFAR10, the improvement of using SpectReg (0.03%) is negligible."
> Indeed, unlike the MNIST case, SpectReg does not help our network on CIFAR-10. We do not claim that it does, the SpectReg number is only provided for reference. However, DataGrad (double backpropagation) does bring an improvement of 0.43% on this dataset. DataGrad is an old (some would say forgotten) technique, but to our knowledge, our paper is the first where it was used in tandem with modern deep networks, and a classification accuracy improvement was demonstrated. We fully agree that repeated runs are the proper way to establish the significance of the CIFAR-10 improvement. In the meantime please see Table 5 of the longer arxiv version of our paper http://arxiv.org/abs/1712.09936, which shows a quite clear trend in our opinion.

---

> > ### Comment · AnonReviewer2 · 2018-03-26
> > **Reference**
> >
> > The reference regarding to my comment "a vanilla CNN can achieve 1.19% error rate on MNIST whereas yours is 3.01%" is:
> >
> > http://rodrigob.github.io/are_we_there_yet/build/classification_datasets_results.html#4d4e495354.
> >
> > See the fifth entry from the bottom.

---

> > > ### Author Response · Authors · 2018-03-31
> > > **review is based on a misapprehension of our claims**
> > >
> > > Training on 2000 data points without data augmentation is a different, harder task than training on 60000 data points with heavy data augmentation, and it is meaningless to compare the error rate between these two very different tasks. Obviously, our baseline system achieves error rate below 1% when trained on 50000 MNIST data points, even without data augmentation.
> > >
> > > We strongly believe that improving the accuracy of deep learning models in the small data regime is an important goal, considering the well-known limitations of current deep learning models in this regime. That’s where we have focused our efforts.

---

### Official Review · AnonReviewer1 · 2018-03-09
**elegant simple idea, incomplete experiments**

**Rating:** 5
**Confidence:** 4

**Review:**

The paper presents two idea on gradient regularization, one old (double backprop) and one new (spectral regzn of the Jac). Spectral regularization is a neat idea. Experiments on MNIST show it makes a marginal improvement on small data regimes. There is also a CIFAR experiment, which appears (?) to be on the whole dataset.  The results are far from conclusive. A stronger case would be made with something like Fig 1 for CIFAR.

---

### Decision · Program_Chairs · 2018-03-20
**ICLR 2018 Workshop Acceptance Decision**

**Decision:**

Reject

**Comment:**

Based on the reviews, this paper has not been accepted for presentation at the ICLR workshop. However, the conversation and updates can continue to appear here on OpenReview.